# Effects of EOs vs. Antibiotics on *E. coli* Strains Isolated from Drinking Waters of Grazing Animals in the Upper Molise Region, Italy

**DOI:** 10.3390/molecules27238177

**Published:** 2022-11-24

**Authors:** Chiara Aquilano, Ligia Baccari, Claudio Caprari, Fabio Divino, Francesca Fantasma, Gabriella Saviano, Giancarlo Ranalli

**Affiliations:** Department of Biosciences and Territory, University of Molise, C. da Fonte Lappone snc, 86090 Pesche, IS, Italy

**Keywords:** essential oil, antibiotic, *E. coli*, drinking water, grazing animal

## Abstract

The health and safety of grazing animals was the subject of microbiological monitoring on natural source of drinking waters in the upper Molise region, Italy. Surface water samples, on spring-summer season, were collected and submitted to analyses using sterile membrane filtration, cultural medium, and incubation. The level of environmental microbial contamination (Total viable microbial count, yeasts and fungi) and faecal presence (Total and faecal coliforms, *E. coli*, and *Salmonellae* spp.) were carried out. By the selective microbiological screening, twenty-three *E. coli* strains from drinking waters were isolated and submitted to further studies to evaluate antibiotic resistance by antibiograms vs. three animal and two diffuse human antibiotics. Furthermore, after a fine chemical characterization by GC and GC-MS, three Essential Oils (EOs) of aromatic plants (*Timus vulgaris*, *Melaleuca alternifolia*, *Cinnamomun verum*) aromatograms were performed and results statistically compared. The effects of EOs vs. antibiotics on *E. coli* strains isolated from drinking waters showed a total absence of microbial resistance. In our experimental conditions, even if some suggestions will be further adopted for better managements of grazing animals, because the health and safety represent a guarantee for both animals and humans.

## 1. Introduction

Molise is the second smallest region of Italy. Its 4438 km^2^ are divided into 55.3% mountainous areas and 44.7% hilly areas. Thanks to its geography, Molise is a region whose subsistence has always been based on breeding and agriculture. In Molise, pasturing is still very commonly practiced. In the vast pastures of upper Molise, freely grazing herds can still be seen. In these places, the animals can eat fresh herbs and quench their thirst in ancient troughs. The practice of mountain pasturing resonates so strongly in Molise that even the ancient Tratturi are still active [1].

The Tratturi are grassy roads by which transhumance takes place, that is, the movement between winter and summer pastures. The Tratturi and transhumance are closely linked: men and flocks in search of food, tracks, and connecting itineraries (Figure 1). The echo of this ancestral tradition has resounded from the pages of numerous international newspapers, such as the New York Times and National Geographic, to say nothing of transhumance’s declaration as a UNESCO cultural heritage in 2019 [2].

Mountain pasturing is a very ancient technique practiced by farmer-ranchers with the aim of exploiting the mountain’s forage resources in the best possible way. The pasture was at the center of a system based on continuous mobility: it was the animal that was brought to forage and not vice versa, as is the case today [3].

The ascent of the herds to the mountain pastures had two main purposes. On the one hand, the small pastures near the cities were freed and could be used as agricultural fields to produce spare fodder for the winter. On the other hand, in the mountains the cattle ate aromatic herbs typical of the high mountain pastures, tastier and nutritious, which improved the quality of the milk produced precisely due to the intrinsic properties of those herbs in promoting lactation, with subsequent benefits for dairy production [4,5].

Another important and nonnegligible aspect of the ascent of the flocks to the alp is the movement that the animals performed, which was essential to firm the meat and improve its quality. With the advent of intensive farming, the practice of grazing has been gradually abandoned. In recent years, however, with the growing awareness of the situations in which animals find themselves in intensive farming, there is a push towards a return to the ancient farming practices in an effort to restore dignity to the animals and also improve the quality of products of animal origin [6,7,8].

Animal welfare is an important reason for providing free grazing and climbing to the mountain pastures during the summer months. In order to verify the quality of resources before leading the flocks to a pasture, periodic analysis on food and water resources have to be carried out to verify the absence of potentially toxic or pathogenic contaminants [9].

A microbiological analysis and a chemical analysis should be carried out once a year to check the status of the water and avoid microbiological problems that could negatively affect the health of the animals and the quality of animal products. Some factors such as inappropriate and incorrect prescription of antibiotics, their overuse in clinics and farms, as well as poor water sanitation in developing countries have exacerbated the phenomenon of antibiotic resistance [10,11,12]. 

In addressing the growing pipeline of antibiotics, natural products such as essential oils (Eos) have been investigated as a promising alternative for controlling the viability of microbial species [13]. Among the main components of EO, there are two distinct chemical groups of biosynthetic origin: terpenes and terpenoids. These compounds are usually responsible for the antimicrobial action against disease-causing bacteria [14,15].

The purpose of this work can be summarized as follow: (a) Microbiological evaluation of the water for livestock. Great attention was focused on the search for microorganisms indicative of contamination (total microbial load at 22 and 37 °C, total coliforms, fecal coliforms, *Escherichia coli*, *Enterococci* spp., and *Salmonella* spp.); (b) isolation of *E. coli* colonies at 44 °C; (c) evaluation of the resistance to different antibiotics (for human and veterinary use) and three essential oils (cinnamon, thyme, and tea tree) of *E. coli* strains isolated in water samples taken from stone pools and stone drinking troughs located in the pastures of upper Molise, Italy (Frosolone, Carpinone, and Civitanova del Sannio). The screenings were carried out by antibiogram and aromatogram tests.

## 2. Results

### 2.1. Chemical Composition of Selected Commercial Oils

The chemical composition of three commercial Eos, cinnamon (*C. verum)*, thyme (*T. vulgaris*), and tea tree (*M. alternifolia*), was analyzed by GC and GC-MS (Figure 2). Appendix A report EOs composition (%). Twenty-two compounds were identified in cinnamon oil, corresponding to 99.8% of the total area. Oxygenated derivatives represent the most abundant class, with 77.14% (±0.30) corresponding to *trans*-cinnamaldehyde and 18.76% (±0.23) corresponding to Eugenol. Twenty-four compounds were identified in Thyme oil, corresponding to 99.7% of the total area. In this commercial extract, the monocyclic monoterpenes and the oxygenate monocyclic monoterpenes were mainly present. Thymol was the most abundant compound (38% ± 0.17), followed by limonene (22.12% ± 1.53) and *p*-cymene (21.83% ± 1.16) from which the thymol was derived; in addition, the isomer carvacrol was also present (8.7% ± 0.14). The composition of commercial tea tree oil shows 24 compounds that were identified, corresponding to 99.2% of the total area. In this commercial extract, the monocyclic monoterpenes and the oxygenate monocyclic monoterpenes were mainly present. Terpinen-4-ol was the most abundant compound (32.52% ± 0.64), followed by terpinene <γ-> (16.52% ± 0.46) and pinene <α-> (11.52% ± 0.38). Hydrodistillated essential oil from *Lavandula angustifolia* L. (season 2020) was previously analyzed and characterized; sixty-one compounds were identified, corresponding to 98.1% of the total area of the sample [16]. Table 1 shows the main chemical compounds of the commercial analyzed EOs compared to lavender natural extracted oil.

Commercial EOs’ composition is quite similar to the natural extracts reported in the literature. *C. verum* exhibit a very high concentration of *trans*-cinnamaldehyde (>70%) and a consistent concentration of eugenol [17,18]. In addition, its antibacterial activity is widely reported in the literature [19]. Our commercial thyme (see Table 1) fit into the “thymol chemotype”, which contains 47% thymol and 20.1% *p*-cymene [20]. Its potential biological activity was widely reported [21,22]. Due its variability, tea tree oil is subject to international regulation ISO 4730:2017 [23]. Our commercial EO composition (Appendix A) was found to be in accordance with this standard [24].

### 2.2. Water Sampling Points

Sampling points were selected following surveys to trace stone drinking troughs and water pools used by herds. Four stone drinking troughs and four water pools were identified and sampled. Sampling points were in the province of Isernia (Table 2, Figure 3).

### 2.3. Environmental Data of Sites Sampled

Environmental temperatures during the sampling campaign in the upper Molise region ranged from 11 °C to 34 °C (night and day) with an average of 22 °C; air humidity ranged from 75% to 90%, and there were no significant instances of rain (less than 10 mm/m^2^), so a marked dry season was observed. The number of hours of sunshine ranged between 10 and 14, with high level of UV radiation not less than 7 h.

### 2.4. Microbial Count in Water Pools and Stone Drinking Troughs

Microbial count results in water pools and stone drinking troughs, analyzed separately, are reported in Figure 4. 

Data about the comparison between the microbial counts in water pools and stone drinking troughs, in the eight samples separately analyzed in triplicate, are shown in Table 3. 

Table 3 and Figure 5 show a high concentration of total microbial count at 37 °C and of coliforms in the water pools, while in the stone drinking troughs the load is markedly lower, with a statistically significant variation. Fecal coliforms are in greater quantities in the water pools, while in stone drinking troughs the microbial concentration is markedly lower; this variation is statistically significant. *Enterococci* spp. have the same tendency as fecal coliforms, only that the difference is significantly higher. The microbial concentration in stone drinking troughs is significantly lower (100×). *Salmonellae* spp. test was negative in all analyzed samples.

### 2.5. Antibacterial Activity against Selected E. coli

The susceptibility to antibiotics of 23 strains of *E. coli* isolated from samples taken from stone drinking troughs and water wells located in pastures in Alto Molise was determined. For each bacterial isolate, sensitivity was evaluated against five antibiotics selected from those usually used both in human medicine and in cattle veterinary medicine. The sensitivity tests were carried out using the diffusion method on an agarized Petri dish (Figure 6). In the evaluation of the effectiveness of the antibiotics used, however, in packaging with synergies of several active ingredients and excipients, and not individual active ingredients, the Manual M100 Performance Standards for Antimicrobial Susceptibility Testing 32th Edition of 2022 was adopted [25], as shown in Table 4 and Table 5 below.

Five antibiotics tested by means of the agar diffusion method showed a variable degree of antimicrobial activity against the 23 strains of *E. coli*, as reported in Table 4 and Table 5. Baytril and Macramid veterinary antibiotics showed the highest antimicrobial activity, remaining active at 1/1000 of the regular dose. Repen and Augmentin are less active against isolated *E. coli* strains, where some of the isolated bacteria are resistant starting at 1/100 of regular dose. Apparently, Bimixin showed a lower efficacy against the tested *E. coli*. Some of the isolated bacteria show resistance or intermediate activity at the regular dose.

### 2.6. Antibacterial Activity of EOs against Selected E. coli

The aromatogram evaluates the sensitivity of a bacterial species to certain EOs. The antimicrobial capacity of three EOs (thyme, cinnamon, tea tree) was evaluated on 23 strains of *E. coli* isolated from samples taken from both stone drinking troughs and water pools located in pastures in Alto Molise. Results are shown in Table 6 and Figure 7. 

Thyme, cinnamon, and tea tree EOs showed a broad spectrum of activity against the Gram-negative strains tested, indicating a strain-dependent sensitivity, and gave the best results, with inhibition zones over 45 mm diameter in some cases. The tea tree EO seems to have the weakest activity against all strain tested. 

In Table 7, at the maximum volumes (20 µL) the activities of thyme and cinnamon EOs were greater than tested antibiotics (both veterinarian and human use). Tea tree oil showed a higher activity only on Macramid and Bimixin. The activity of tea tree EO at 2 µL turned out to be significantly lower than all tested antibiotics; furthermore, thyme EO was better only when compared to Macramid and Bimixin, while cinnamon EO appeared to have the greatest activity against all tested antibiotics.

Figure 7a shows an example of aromatograms for EOs on *E. coli* isolated strains after 48 h of incubation. Under our experimental conditions, after cinnamon EO addition (20 µL) we noted a deep alteration on the bottom of the Petri dish. It was a hole with the same dimension as the well (Figure 7b). A comparison test among benzaldehyde, formaldehyde, formamide, and cinnamon EO confirmed these results (Figure 7c). This Petri dish alteration could be related to the high level (77.14%, see Table 1) of cinnamaldehyde <(E)>. This effect was not sporadic, and it was observed several times when cinnamon EO was added at higher volume. 

Furthermore, in another test, when 1.0 mL cinnamon EO was added on a paper filter inside an empty Petri dish (PE-LD 04 CE), a complete hard adhesion between the lid and the dish was observed. The Petri dish showed a loss of transparency, opacity, deformations, and a depression area on the surface (Figure 7d), with the consequence that the plate could not be opened.

## 3. Discussion

Water can be a vehicle for infections, and for this reason, it is one of the most important parameters to be kept under control for farm management. In case of infections and disturbances caused by the ingestion of water containing high concentrations of pathogenic microorganisms, the use of antibiotics and antimicrobials is required. Antibiotics are effective drugs that play an important role in treating infections in both human and animal environments. Antibiotic resistance develops naturally but is accelerated when antibiotics are used incorrectly in both humans and animals [27]. The microbiological indicators for grazing water quality, highlighted in Table 3, such as the total microbial load at 22 and 37 °C, levels of total coliforms, fecal coliforms, *E. coli* at 44 °C, and enterococci are elevated and non-compliant when compared to human consumption standards [28].

Twenty-three *E. coli* isolates show a high sensitivity to antibiotics currently most used in veterinary medicine (Repen, Macramid, Baytril), but only when tested at the highest concentrations. *E. coli* was also found to be sensitive to antibiotics for human use. Specifically, Augmentin was found to be efficient even at the lowest concentrations, but Bimixin showed good efficacy only when used at the maximum dose (see Table 7). From our point of view, these results must be considered of great importance, because they confirm the absence of antibiotic resistance in environmental *E. coli* strains. It is an important result, because these bacteria are identified as fecal indicators of the water quality of grazing animals in the upper Molise.

However, the inadequacy of some antibiotics (Bimixin and Repen in this study) could be considered a potential alarm bell indicating the onset of future resistant microbial forms [29,30]. One of the factors that may have favored the selection of resistant bacteria within the farm is the repeated use of antibiotic therapies. Among the objectives of the plans to combat basic antibiotic resistance is the reduction of the use of drugs in the livestock sector, especially if used improperly, as an alternative to compliance with good farming practices and company biosecurity [31,32]. 

By observing the results obtained in Table 6, it can be noted that among the EOs used, only cinnamon has an excellent yield in terms of antibacterial capacity. Therefore, among the oils used, only cinnamon EO shows a large halo of inhibition (diameter > 18 mm) with the smallest volume used. Cinnamon is plant that grows in Asian tropical countries. Depending on the raw material, bark, or cinnamon leaf, the concentration of *trans*-cinnamaldehyde and eugenol may vary [33].

Based on the results in Table 1, *trans*-cinnamaldehyde and eugenol are 77.14% and 18.76%, respectively, so commercial EO should be derived from cinnamon bark. Our results on the well diffusion assay (Table 6) fully agree with data in the literature, where the cinnamon EO shows an antimicrobial activity against *Staphylococcus aureus*, *E. coli*, *Acinetobacter baumannii*, and *Pseudomonas aeruginosa*, compared to tea tree oil and other EOs [34]. High concentrations of cinnamaldehyde are presumably also responsible for the melting spot observed on the Petri dish (Figure 7b,d). No one major tested component gave same result (Figure 7c). Therefore, the use of this EO must be properly considered both to avoid harm for users and to enhance the potential it offers.

Although extensive studies have been conducted to explore the antimicrobial potential of EO, the results of several studies do not fully agree in terms of EOs’ efficiency in killing microbes or mitigating their stability under various environmental conditions, such as in the presence of salts. There are also limited studies on the relationship between the structure of EO compounds related to antimicrobial activity [35]. However, the mechanisms by which the EO function are not fully understood in detail. Furthermore, negative impacts of excessive doses of EO on human health have been reported [36].

Although generally considered safe, natural products and EOs are not completely safe due to the diversity of the metabolites they contain when used as antimicrobial agents [37]. The large number of secondary metabolites can have synergistic or antagonistic properties when used in non-standardized form. Therefore, it is necessary to optimize the analysis of a range of EOs and compounds more clearly, for effective but safe use of EOs for humans. The formation of residues in biological systems and allergic reactions associated with the use of EOs are also among other challenges related to their use as antimicrobial agents [38].

Although a multiplicity of EOs are available, only a few have gained approval as a preservative in food. This is because some food components drastically reduce the antimicrobial efficiency of EOs. Consequently, it is desirable to develop more effective and validated food model systems that have a close resemblance to food components. This will predict the effect of food on EOs’ antimicrobial potentials and help in the optimization of EO as food preservatives. Stability is another issue associated with the use of EOs due to their thermolabile nature. EOs are generally volatile at normal temperatures and should be stored in a cool, dry environment. Exposure to high temperatures and humidity could cause them to decompose and thus reduce their effectiveness as antimicrobial agents [39,40].

Natural products must be carefully evaluated for the antimicrobial activity of their purified EOs before describing them as effective or ineffective antimicrobial agents. EOs can be used in conjunction with common herbs such as spices, which will improve their storage capacity for various foods including fish, sauces, meat, and soups. 

As previously mentioned, EOs alone can offer only a mild antimicrobial spectrum, which leads to an ineffective organoleptic profile for users. Consequently, research on the combined synergic effect of EOs and aromatic herbs could be designed in the future for a more effective and optimized use of these oils as preservatives while preserving their aroma and spicy taste. 

The effects of tested EOs, in comparison to some veterinarian and human antibiotics, (Table 7) provide promising and significant potential use for the development of alternative, substitutive therapeutics and treatment for infectious diseases caused by multidrug-resistant microorganisms, like by *E. coli* strains, not excluding their combination and synergic actions [41]. 

Their use in open environments may not bet economically or ecologically sustainable. Therefore, it is necessary to adopt sanitizing protocols for stone drinking troughs; in addition, there is a need for periodic checks on the quality of water intended for livestock, in order to avoid the onset of epidemics due to the ingestion of unsuitable water. For drinking pools, it is difficult to carry out sanitation and disinfestation plans; therefore, based on the data collected, these natural water sources are unsuitable for supply and administration to herds in the pastures. A critical evaluation of water sources and managerial adjustments to create ideal watering sources may result in improved performance and profits [42].

## 4. Materials and Methods

### 4.1. Essential Oils 

Three EOs (cinnamon, thyme, tea tree) were supplied by a local store (Centro Erboristico Aldebaran sas, Pescara, Italy; manufactured by Esperis S.p.A., Milano, Italy). Lavender EO was already hydrodistillated and previously characterized [16].

### 4.2. GC-FID Analysis

The characterization of the three essential oil samples was determined with a gas chromatography system GC 86.10 Expander (Dani) equipped with an FID detector, an Rtx^®^-5 Restek capillary column (30 m × 0.25 mm i.d. 0.25 um film thickness) (diphenyl-dimethyl polysiloxane), a split/splitless injector heated to 250 °C, and a flame ionization detector (FID) heated to 280 °C. The column temperature was maintained at 40 °C for 5 min, then programmed to increase to 250 °C at a rate of 3 °C/min and held using an isothermal process for 10 min; the carrier gas was He (1.0 mL/min); 1 µL of each sample was dissolved in n-hexane (1:500) and injected. The experiment was repeated three times.

### 4.3. GC/MS Analysis

The GC-MS analyses were performed on a Trace GC Ultra (Thermo Fisher Scientific) gas chromatography instrument equipped with an Rtx^®^-5 Restek capillary column (30 m × 0.25 mm i.d. 0.25 µm film thickness) and coupled with an ion-trap (IT) mass spectrometry (MS) detector Polaris Q (Thermo Fisher Scientific, Waltham, MA, USA). A Programmed Temperature Vaporizer (PTV) injector and a PC with a chromatography station Xcalibur (Thermo Fisher Scientific) was used. The ionization voltage was 70 eV; the source temperature was 250 °C; full scan acquisition in positive chemical ionization was from *m*/*z* 40 up to 400 a.m.u. at 0.43 scan s^−1^. The GC conditions were the same as those described above for the gas chromatography (GC-FID) analysis.

### 4.4. Identification of Essential Oil Components

The identification of the essential oil components was based on the comparison of their Kovats retention indices (KIs) and LRI (linear retention indices). These indices were determined in relation to the t_R_ values of a homologous series of n-alkanes (C8–C20) injected under the same operating conditions [43,44].

The MS fragmentation patterns of a single compound were those from the NIST 02. Adams and Wiley 275 mass spectral libraries [45,46]. The relative contents (percentage, %) of the sample components were computed as the average of the GC peak areas obtained in triplicate without any corrections [47].

### 4.5. Sampling and Culture Media

Four stone drinking troughs and four water pools were sampled. Samples were collected following specific guidelines [48]. Carefully sterilized glass bottles were used for sampling. Water was filtered on membrane (MF), counting the colonies developed on membrane [CFU/100 mL = n. colonies counted/100 mL of sample filtered [49]. The procedure provides an asepsis condition and a cellulose membrane of 0.45 μm (Sartorius Italy s.r.l.). Finally, the cellulose filter is transferred to a suitable agarized culture medium in Petri dishes. Tests were conducted separately, in triplicate.

PCA agar (DIFCO) was used for microbial count; C-EC agar (Biolife Italiana), which discriminates between coliforms and *E. coli*; Slanetz and Bartley (Merck) that allows to discriminate enterococci; Hektoen Enteric Agar (Biolife Italiana), for the determination of *Salmonellae* spp. at 37 °C. Yeast extract agar (Biolife Italiana) was used for the determination of the total microbial count at 37.5 °C and 22 °C, in all types of water in accordance with the recommendations [50]. Sabouraud Dextrose Agar (Biolife Italiana) was used for the isolation, cultivation, and maintenance of non-pathogenic and pathogenic species of fungi and yeasts. 

### 4.6. Antibiotics and Antibiogram 

For the antibiogram were selected three antibiotics frequently used in veterinary medicine for the prevention and treatment of pathological conditions in cattle farms, as well as two antibiotics for human use. The first veterinary antibiotic is Repen (Fatro Industria Farmaceutica Veterinaria S.p.A., Italy; Agrifarmavet, Isernia, Italy), an association of penicillin and dihydrotreptomycin, composition 1 mL benzyl penicillin 200,000 U.I., dihydrotreptomycin sulfate 250 mg/mL. The second antibiotic is Macramid (Elanco Italia S.p.A., Italy; Agrifarmavet, Isernia, Italy), which is used for systemic applications, and it contains lincomycin/spectinomycin as active ingredients. The composition includes lincomycin hydrochloride 56.7 mg/mL equal to lincomycin 50 mg/mL and spectinomycin sulphate tetrahydrate 150 mg/mL equal to spectinomycin 100 mg/mL. The third veterinary antibiotic is Baytril 10% (Elanco Italia S.p.A., Italy; Agrifarmavet, Isernia, Italy), which is used for systemic treatments. It contains Enrofloxacin 100 mg/mL in each dose. The antibiotics for human use were Augmentin (Glaxo Smith Kline S.p.A., Verona, Italy), composed of amoxicillin 875 mg/mL and clavulanic acid 125 mg/mL, and Bimixin (Sanofi Italia S.p.A., Milano Italy), composed of neomycin sulfate 33.11 mg/mL and bacitracin 33.78 mg/mL. Previously isolated *E. coli* strains were suspended in 10 mL of sterile nutrient broth culture medium and incubated at 44 °C for 24 h. After evaluation of the turbidity at the spectrophotometer, 300 µL of bacterial suspension was collected and distributed evenly, by a sterile L-loop, on the surface of each Petri dish containing 15 mL of plant count agar culture medium (PCA). Since liquid antibiotics were available, it was decided to carry out the antibiogram exclusively using the well diffusion technique, as described by Balouiri et al. [51]. 

Antibiotic response: N.D. = Not Diluted standard dose, 10^−x^ = dilution factor of standard dose. According to the agar well diffusion assay, we summarize: Weak activity (W) = inhibition halo ≤13 mm; intermediate activity (I) = inhibition halo between ≥13.1 mm and ≤17.9 mm; strong activity (S) = inhibition halo ≥18.0 mm [26].

Each well, prepared by a sterile glass Pasteur pipette, was filled with 20 µL of antibiotic solutions and their dilutions as follows: Repen (ranging from ND = 3.93 mg benzyl penicillin and 5 mg dihydrostreptomycin to 0.39 µg (10^−4^ dilution) and 0.5 µg (10^−4^ dilution) benzylpenicillin and dihydrostreptomycin, respectively); Macramid (ranging from ND = 1.0 mg lincomycin and 2.0 mg spectinomycin to 0.1 µg lincomycin and 0.2 µg spectinomycin, respectively; Baytril (ranging from ND = 2.0 mg to 0.2 µg of enrofloxacin); Augmentin (ranging from ND = 17.5 mg amoxicillin and 2.5 mg clavulanic acid to 1.75 µg and 0.25 µg amoxicillin and clavulanic acid, respectively); Bimixin (ranging from ND = 0.66 mg neomycin sulfate and 0.67 mg bacitracin to 0.66 µg neomycin sulfate + 0.67 µg bacitracin, respectively), as reported in Table 4 and Table 5. 

### 4.7. Aromatogram

This test was performed in vitro according to Balouiri et al. [51]. Each well, created in the culture medium using a sterile glass Pasteur pipette, was filled with 20 µL of essential oil or dilutions or diluent as control. Tween 20 (Sigma-Aldrich, St Louis, MO, USA) was used a negative control. EO response: N.D. = Not Diluted standard dose, 10^−x^ = dilution factor of standard dose. According to the agar well diffusion assay, we summarize: Weak activity (W) = inhibition halo ≤13 mm; intermediate activity (I) = inhibition halo between ≥13.1 mm and ≤17.9 mm; strong activity (S) = inhibition halo ≥18.0 mm [26].

### 4.8. Statistical Analysis

All the statistical results were expressed in terms of mean ± SD while the evaluation of the significance concerning the several situations was carried out using Student’s *t*-tests, adjusted with the Bonferroni correction when necessary, and using the software ***R*** www.r-project.org (accessed on 22 September 2022); the significance level was set at *p* < 0.05 and *p* < 0.01.

The comparisons between Antibiotics and EOs were carried out using data sets (20 µL and 2 µL of EOs and antibiotics) when antibiotic dilution showed the first inhibition toward 23 *E. coli* isolates. 

### 4.9. Environmental Parameters of Sites Sampled

Environmental data were collected at the sampled sites within the time frame covered by the microbiological surveys: daily temperatures (low–high, average) and moisture were constantly monitored on-site by portable data loggers (RHiLog Escort Data Loggin System Ltd., Auckland, New Zealand). Furthermore, rainfall (mm/m^2^) and number of hours of sunshine were provided by the local ARPA environmental monitoring station, Isernia, Italy.

## 5. Conclusions

In our experimental conditions, the effects of EOs vs. antibiotics on *E. coli* strains isolated from drinking water showed a total absence of microbial resistance. From our point of view, these results are of great importance because they confirm the absence of antibiotic resistance in 23 environmental *E. coli* strains isolated and identified as fecal indicators of water quality of grazing animals in the upper Molise region. Our results can be used to adopt suggestions for better management of grazing animals:
(i)provision should be made for the establishment of watering points that are safer and that can be controlled from a health point of view; (ii)the municipalities should provide for the positioning of stone drinking troughs supplied with water from the municipal water pipes; (iii)in order to assess the effects of climate change (presence/absence of rain, temporal distribution, variations in ambient water temperature, UV exposure, leaching of salts from the soil to the watering hole, metal content, dilution/concentration of viable microorganisms, survival dynamics, variations in microbial communities, etc.), it is advisable to carry out further studies over a longer time span (i.e., 5–8 years) on the quality of drinking water for free-range animals.

These facilities will reflect both on the welfare of the animals that use the water and the quality of the final products of animal origin (milk, cheese, etc.). 

Finally, it is advisable to be very careful when handling EOs because of their potential harmfulness due to certain chemical components inside.

## Figures and Tables

**Figure 1 molecules-27-08177-f001:**
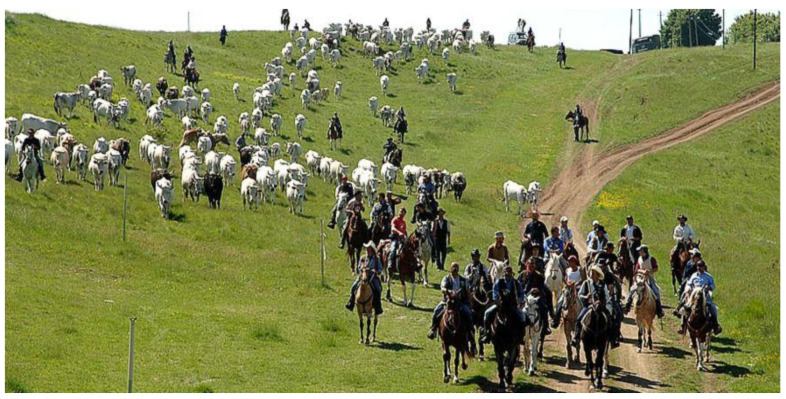
Men, animals, and territory: the three elements of *transumanza*, an ancient traditional type of grazing in upper Molise, Italy, included in the cultural heritage list in 2019 by UNESCO [2].

**Figure 2 molecules-27-08177-f002:**
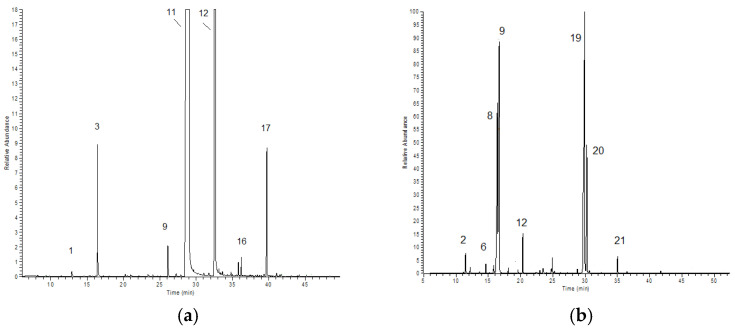
The GC-MS TIC chromatogram of commercial EOs: cinnamon (**a**), thyme (**b**), tea tree (**c**). Numbers refer to those reported in Appendix A.

**Figure 3 molecules-27-08177-f003:**
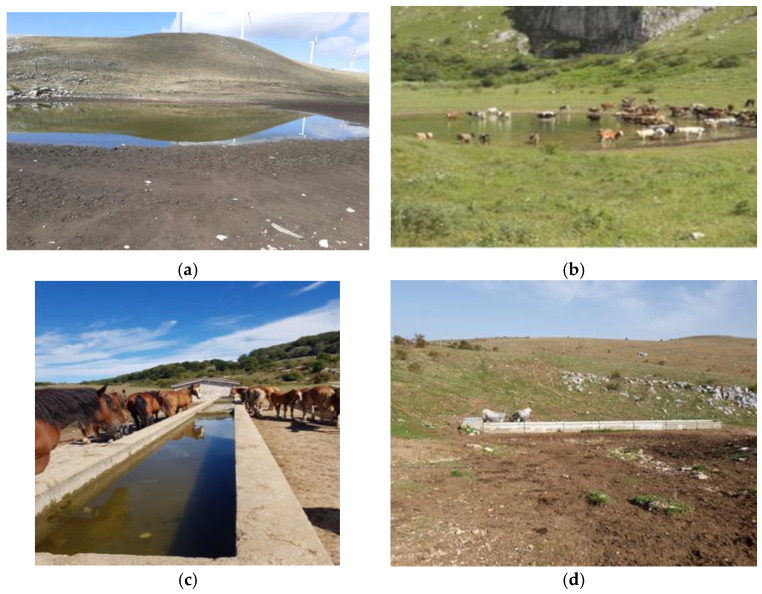
Examples of water pools and stone drinking troughs used by herds sampled in this study. (**a**) Lago di Acquaspruzza—P4, (**b**) Lago dei Castrati—P2, (**c**) Carpinone (IS)—A2, (**d**) Frosolone (IS)—A7.

**Figure 4 molecules-27-08177-f004:**
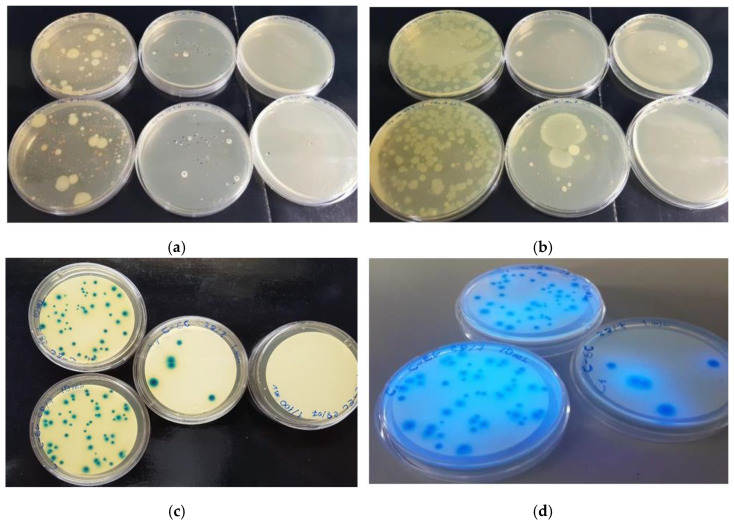
Example of microbial count on samples coming from water pools (A1). Dilutions or filter were plated on PCA agar plate for total microbial count at 37 °C (**a**), PCA agar plate for total microbial count at 22 °C (**b**), C-EC agar plate for fecal coliforms 44 °C (**c**), C-EC agar plate *E. coli* at 44 °C (**d**).

**Figure 5 molecules-27-08177-f005:**
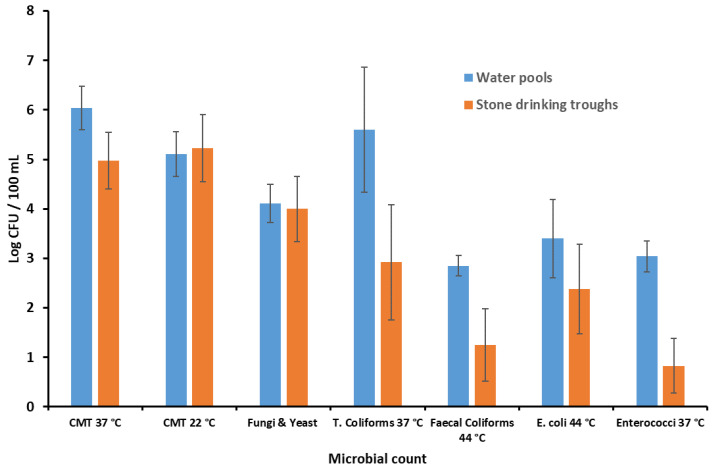
Comparison between a total of eight samples from water pools and stone drinking troughs, separately analyzed, in triplicate. Results are expressed as the mean ± SD.

**Figure 6 molecules-27-08177-f006:**
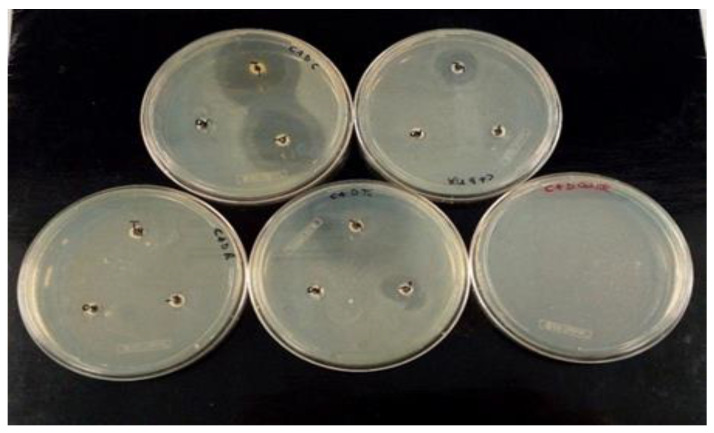
Example of sensitivity/resistance of *E. coli* isolated strains against the Repen antibiotic on Petri dishes after 48 h of incubation.

**Figure 7 molecules-27-08177-f007:**
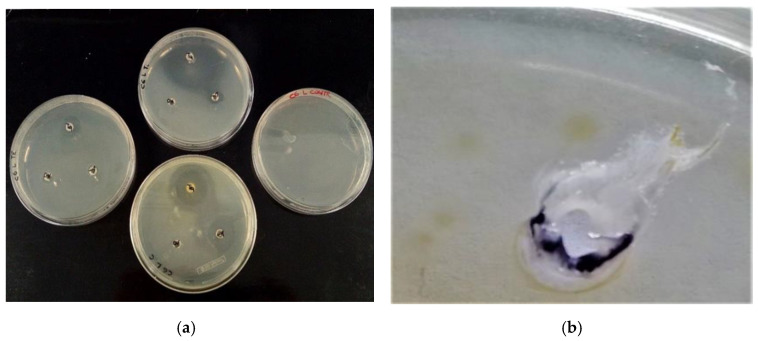
Example of EOs’ aromatograms on sensitivity/resistance of *E. coli* isolated strains on Petri dishes (**a**); Alteration on plastic material on Petri dish surface after 48 h by addition of 20 µL of cinnamon EO (**b**); Comparison test on Petri dish between same amount (20 µL) of benzaldehyde (1), formaldehyde (2), formamide (3), and cinnamon EO (4) (**c**); Alteration of the plastic material of the lid of a Petri dish after adding 1 mL of cinnamon EO (**d**).

**Table 1 molecules-27-08177-t001:** Main chemical composition (>1.0%) of the three commercial EOs (cinnamon, thyme, tea tree) compared to *L. angustifolia* natural extract. Lavender EO was previously characterized [16].

N.	Compound	Area % ± SD			
	**EOs**	**Cinnamon**	**Thyme**	**Tea tree**	**Lavender**
1	Limonene	1.35 ± 0.02	22.12 ± 1.53	2.25 ± 0.35	
2	Cinnamaldehyde <(E)>	77.14 ± 0.30			
3	Eugenol	18.76 ± 0.23			
4	Methoxy cinnamaldehyde <(E)-*o*->	1.36 ± 0.04			
5	Pinene <α->		1.33 ± 0.02	11.52 ± 0.38	
6	Cymene <p->		21.83 ± 1.16	5.01 ± 0.23	
7	Linalool		2.47 ± 0.02		36.0 ± 0.01
8	Thymol		38.05 ± 0.17		
9	Carvacrol		8.70 ± 0.14		
10	Pinene <β->			4.24 ± 0.16	
11	Terpinene <α->			8.15± 0.22	
12	Cineole <1,8->			3.84 ± 0.31	9.00 ± 0.3
13	Terpinene <γ->			16.52 ± 0.46	
14	Terpinolene			2.50 ± 0.13	
15	Terpinen-4-ol			32.52 ± 0.64	6.81 ± 0.02
16	Terpineol <α->			4.98 ± 0.08	
17	Guaiene <α->			3.24 ± 0.05	
18	Valencene			1.79 ± 0.01	
19	Camphor				6.80 ± 0.07
20	Borneol				19.4 ± 0.1
21	Linalyl acetate				2.75 ± 0.02
22	Farnesene <β->				1.50 ± 0.01

**Table 2 molecules-27-08177-t002:** Type, name, location coordinate of the site, and code of sampling points.

Type	Name	Location Coordinate	Code
Water pool	Lago di Carpinone (IS)	41.6100140 N 14.3753190 E	P1
Water pool	Lago Castrati (IS)	41.6153035 N 14.3991883 E	P2
Water pool	Lago delle Cannavine (IS)	41.6053227 N 14.4131814 E	P3
Water pool	Lago di Acquaspruzza (IS)	41.5947320 N 14.4044120 E	P4
Stone trough	Sessano del Molise (IS)	41.6337403 N 14.3301754 E	A1
Stone trough	Carpinone (IS)	41.5947320 N 14.4044120 E	A2
Stone trough	Frosolone (IS)	41.6137290 N 14.3975582 E	A3
Stone trough	Frosolone (IS)	41.6096087 N 14.3975582 E	A4

**Table 3 molecules-27-08177-t003:** Comparison of microbial count in water pools and stone drinking troughs samples. Results expressed as the mean of three replicates separately analyzed with ±SD. ns: not significant; *p* < 0.05 and *p* < 0.01 significant difference.

Microbial Group	Water Pools	Stone Drinking Troughs	*p*
	Log CFU/100 mL (±SD)	Log CFU/100 mL (±SD)	
TMC 37 °C	6.04 (0.44)	4.97 (0.57)	<0.02
TMC 22 °C	5.11 (0.45)	5.23 (0.68)	ns
Fungi & Yeast 22 °C	4.11 (0.39)	4.0 (0.66)	ns
Tot Colif 37 °C	5.60 (1.26)	2.92 (1.16)	<0.05
Faecal Colif 44 °C	2.85 (0.21)	1.25 (0.73)	<0.05
*E. coli* 44 °C	3.40 (0.79)	2.38 (0.91)	ns
Enterococci 37 °C	3.04 (0.31)	0.83 (0.55)	<0.01
*Salmonellae* spp. 37 °C	absent	absent	ns

**Table 4 molecules-27-08177-t004:** Antimicrobial activity of three antibiotics commonly used on cattle with respect to 23 *E. coli* isolates. Data interpretation in Section 4.6. Here are reported the mean halo diameter calculated for each column ± SD according to Mazzarrino et al. [26].

*E. coli* Isolates	REPEN [Benzylpenicillin 196.66 mg/mL + Dihydrostreptomycin 250 mg/mL]				MACRAMID [Lincomycin 50 mg/mL + Spectinomycin 100 mg/mL]			BAYTRIL [Enrofloxacin 100 mg/mL]		
	N.D.	10^−2^	10^−3^	10^−4^	N.D.	10^−3^	10^−4^	N.D.	10^−3^	10^−4^
Activity	S	S	W	W	S	W	W	S	S	I
Mean	32.6	18.2	11.2	2.0	36.1	7.5	0	47.2	20.3	16.0
(±SD)	4.01	5.25	6.37	5.17	4.99	5.03	0	4.08	4.81	7.39

**Table 5 molecules-27-08177-t005:** Antimicrobial activity of two antibiotics commonly used on humans with respect to 23 *E. coli* isolates Data interpretation in Section 4.6. Here are reported the mean halo diameter calculated for each column ± SD according to Mazzarrino et al. [26].

*E. coli* Isolates	AUGMENTIN [Amoxicillin 875 mg/mL + Clavulanic acid 125 mg/mL]				BIMIXIN [Neomycin Sulfate 33.11 mg/mL + Bacitracin 33.78 mg/mL]			
	N.D.	10^−2^	10^−3^	10^−4^	N.D.	10^−1^	10^−2^	10^−3^
Activity	S	S	W	W	S	W	W	W
Mean	40.8	20.3	12.7	1.10	19.6	11.3	4.2	0
(±SD)	2.92	12.19	9.04	4.03	4.09	2.35	3.48	0

**Table 6 molecules-27-08177-t006:** Antimicrobial activity of three EOs (thyme, cinnamon, tea tree) on 23 strains of *E. coli*. Data interpretation in Section 4.7. Here are reported the mean halo diameter calculated for each column ± SD according to Mazzarrino et al. [26].

*E. coli* Isolates	Essential Oils						
	Thyme		Cinnamon		Tea Tree		Tween 20
	20 µL	2 µL	20 µL	2 µL	20 µL	2 µL	20 µL
Activity	S	W	S	S	I	W	W
Mean (mm)	36.3	12.9	30.3	22.2	15.4	0	0
(±SD)	0.98	0.46	0.64	0.38	0.41	0	0

**Table 7 molecules-27-08177-t007:** Statistical significance among aromatograms of three EOs (thyme, cinnamon, tea tree) at two volumes (20 µL and 2 µL) vs. antibiotics for veterinarian and human use, respectively. Data interpretation in Section 4.8.

Statistical Significance EOs	Veterinary Medicine			Human Use	
	Repen 10^−2^	Macramid 10^−3^	Baytril 10^−3^	Augmentin 10^−2^	Bimixin 10^−1^
Thyme 20 µL	<0.01	<0.01	<0.01	<0.01	<0.01
Cinnamon 20 µL	<0.01	<0.01	<0.01	<0.01	<0.01
Tea tree 20 µL	ns	<0.01	<0.01	ns	<0.01
Thyme 2 µL	<0.01	<0.01	<0.01	ns	<0.05
Cinnamon 2 µL	<0.05	<0.01	ns	ns	<0.01
Tea tree 2 µL	<0.01	<0.01	<0.01	<0.01	<0.01

## Data Availability

Not applicable.

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
