# Peer review of "Effects of EOs vs. Antibiotics on E. coli Strains Isolated from Drinking Waters of Grazing Animals in the Upper Molise Region, Italy"

_molecules, 2022, doi:10.3390/molecules27238177_

Round 1
Reviewer 1 Report
The results must improve.
The tables with the results of antibiotics must be similar to essential oil.
The results must show the statistical analysis with of Chi-square test.
Author Response
Comments and Suggestions for Authors
Thank you very much for your suggestions. In this last version of the manuscript, we followed your indications. All changes on text are reported in red color.
The results must improve. YES, we improved the result section.
The tables with the results of antibiotics must be similar to essential oil.
YES, we agree. We changed the tables with antibiotics; now they are similar to the essential oils. Thanks.
The results must show the statistical analysis with of Chi-square test.
YES. We agree. All results, including antibiotics and EOs, were submitted to new statistical analyses by T-student test. This test is suitable when several groups of data must be compared among them.

Reviewer 2 Report
There are typing errors throughout the text, there are separations (.) that are not necessary, you should rewrite this. In the article I send you the text highlited some points.
Line 104: (value 98.5 %) space is not necessary.
Line 96-102: add the standard deviation in the percentage values.
Figure 1 and 2: improve the quality of the images.
Table 2: Correct the code of location 1 is missing.
Line 141-144: This is methodology.
Figure 4: Choose images that best represent what you want to indicate, is it possible to reduce the number of images?
line 159-160, 178-180, 184-186, 202-204: this goes in the image legend or methodology section.
Table 3. sample 1, p-value is missing
Line 286: n-exane ???
Line 300: rewrite this section (Kovats index KI) (Linear retention index LRI)
Line 358 -372: the conclusion does not provide more information, it should talk about the results obtained and what these results are useful for in the future.
Author Response
Thank you very much for your suggestions. In this last version of the manuscript, we followed your indications. All changes on text are reported in red color.
Comments and Suggestions for Authors
There are typing errors throughout the text, there are separations (.) that are not necessary, you should rewrite this. In the article I send you the text highlighted some points.
YES, we corrected several typing errors in the text, according to the suggestions.
Line 104: (value 98.5 %) space is not necessary. Done.
Line 96-102: add the standard deviation in the percentage values. Done
Figure 1 and 2: improve the quality of the images.
Thank you for your suggestion; we improved the Figures 1 and 2.
Table 2: Correct the code of location 1 is missing. The code was not missing, but in the pdf file of the manuscript the code shifted down…Now we solved the problem. Thanks.
Line 141-144: This is methodology. We agree. Now we moved it in Mat & Meth.
Figure 4: Choose images that best represent what you want to indicate, is it possible to reduce the number of images? Yes, we choose best images and two Figures were omitted.
line 159-160, 178-180, 184-186, 202-204: this goes in the image legend or methodology section.
Yes, we agree. We moved the text on methodology section.
Table 3. sample 1, p-value is missing. Yes, now the p-value is present.
Line 286: n-exane ??? We correct like n-hexane.
Line 300: rewrite this section (Kovats index KI) (Linear retention index LRI).
Yes, we correct indices and we changed the section.
Line 358 -372: the conclusion does not provide more information, it should talk about the results obtained and what these results are useful for in the future.
YES, we agree. The conclusion section was integrated with more information on both the results obtained and the future implications.

Reviewer 3 Report
In this article, there are 8 samples of water: 4 from water pools and 4 stone drinking troughs. It is not clear what samples were analyzed. It seems to be an average sample for each type of water. Please clarify this issue.
Although the authors presented each sampling point's geographic coordinates and environmental conditions, these data were hardly used.
Some notations are necessary in Figure 4 to identify which samples are analyzed. Also, the caption of Figure 4 should provide more details about the analyzed samples. Same thing for Figure 6. In Figure 5 is there an average sample?
More data about the origin of these essential oils should be given. Also, the causal link must be made between the chemical composition of the essential oils and the antimicrobial performance against E coli. I would like to see the article with more data about the aromatograms and images of these tests.
The authors should also provide the composition of Neomycin.
What was the concentration of each antibiotic used in the antibiograms?
Spelling mistakes:
cultural media in Abstract,
n-exane pg 11r aw287
an unfinished sentence Pg 3 raw 94,
The procedure: provides: asepsis condition pg 12 r aw 312
Over the time covered by the microbiological investigations Pg 13 raw352
References 23, 25, and 47must be changed according to the requirements of the journal.
Author Response
Thank you very much for your suggestions. In this last version of the manuscript, we followed your indications. All changes on text are reported in red color.
Comments and Suggestions for Authors
In this article, there are 8 samples of water: 4 from water pools and 4 stone drinking troughs. It is not clear what samples were analyzed. It seems to be an average sample for each type of water. Please clarify this issue. Thank you for your suggestions. Now we reported that all 8 water samples were separately analyzed, in triplicate; then submitted to statistical analyses.
Although the authors presented each sampling point's geographic coordinates and environmental conditions, these data were hardly used. Yes. We agree and we put them in correlation with data reported in § 2.3.
Some notations are necessary in Figure 4 to identify which samples are analyzed. Also, the caption of Figure 4 should provide more details about the analyzed samples. Same thing for Figure 6. In Figure 5 is there an average sample? Yes, we adopted the suggestions received. Now more info are presented in all Figures 4, 5 and 6.
More data about the origin of these essential oils should be given. Also, the causal link must be made between the chemical composition of the essential oils and the antimicrobial performance against E coli. I would like to see the article with more data about the aromatograms and images of these tests. We fully agree with the referee. All this section was improved with more data and images, including the effect of Cinnamon oil on microflora and plastic materials (see Figures 7) and main chemical components (Cinnamaldehyde). Furthermore, all data were submitted to statistical analyses by T-student test.
The authors should also provide the composition of Neomycin. YES, now we report the correct name of antibiotic containing Neomycin: Bimixin (see Table 5 and following….).
What was the concentration of each antibiotic used in the antibiograms?
We reported the concentration of each antibiotic used (see Mat & Meth § 4.6);
Spelling mistakes:
cultural media in Abstract, Done
n-exane pg 11r aw287 Done (n-hexane)
an unfinished sentence Pg. 3 raw 94, Done
The procedure: provides: asepsis condition pg 12 r aw 312 Done
Over the time covered by the microbiological investigations Pg 13 raw352 Corrected
References 23, 25, and 47 must be changed according to the requirements of the journal. Done.

Round 2
Reviewer 1 Report
The authors did the recommendations
Author Response
Dear Reviewer 1, many thanks by all Authors.
Reviewer 2 Report
The article has improved remarkably, but there are some small mistakes:
line 247: Table 7, use the MPDI format
line 362: correct punctuation "Three EOs (Cinnammon, Thyme, Tea Three)".
line 374 : it is not necessary to repeat n-hexane (1:500 n-hexane solution), just use (1:500).
Author Response
Dear Reviewer 3, many thanks for the last suggestions. We agree and in this new text we corrected all details.
line 247: Table 7, use the MPDI format: OK
line 362: correct punctuation "Three EOs (Cinnammon, Thyme, Tea Three)". OK
line 374 : it is not necessary to repeat n-hexane (1:500 n-hexane solution), just use (1:500). OK
Reviewer 3 Report
The authors answered all the requirements from the review report. I recommend publishing the article in its current form.
Author Response
Dear Reviewer 3, many thanks by all Authors.